# Multi-view Contrastive Graph Clustering

**Erlin Pan, Zhao Kang**[*]
School of Computer Science and Engineering,
University of Electronic Science and Technology of China, Chengdu, China
wujisixsix6@gmail.com zkang@uestc.edu.cn

## Abstract

With the explosive growth of information technology, multi-view graph data have become increasingly prevalent and valuable. Most existing multi-view clustering techniques either focus on the scenario of multiple graphs or multi-view attributes. In this paper, we propose a generic framework to cluster multi-view attributed graph data. Specifically, inspired by the success of contrastive learning, we propose multi-view contrastive graph clustering (MCGC) method to learn a consensus graph since the original graph could be noisy or incomplete and is not directly applicable. Our method composes of two key steps: we first filter out the undesirable high-frequency noise while preserving the graph geometric features via graph filtering and obtain a smooth representation of nodes; we then learn a consensus graph regularized by graph contrastive loss. Results on several benchmark datasets show the superiority of our method with respect to state-of-the-art approaches. In particular, our simple approach outperforms existing deep learning-based methods.

## 1 Introduction

An attributed graph contains of node features and edges characterizing the pairwise relations between nodes. It is a natural and efficient representation for many real-world data [Liu et al., 2021]. For example, social network users have their own profiles and the topological graph reflects their social relationships. Different from most classical clustering methods like K-means and hierarchical clustering which only focus on Euclidean data, graph clustering divides unlabeled nodes of graph into clusters. Typical graph clustering methods first learn a good representation of graph and then apply a classical clustering method upon the embeddings. For example, large-scale information network embedding (LINE) [Tang et al., 2015] is a popular graph representation learning method, which can preserve both local and global information and scale up easily to large-scale networks. To incorporate node features and graph structure information, graph autoencoder (GAE) [Kipf and Welling, 2016] employs a graph convolution network (GCN) encoder and achieves significant performance improvement. The real-life data are often collected from various sources or obtained from different extractors, thus are naturally represented by different features or views [Kang et al., 2021, 2020c]. Each view could be noisy and incomplete, but important factors, such as geometry and semantics, tend to be shared among all views. Therefore, features and graphs of different views are complementary, which implies that it's paramount to integrate all features and graphs of diverse views to improve the performance of clustering task.

Numerous graph-based multi-view clustering methods have been developed to capture the consensus information shared by different views in the literature. Graph-based multi-view clustering constructs a graph for each view and fuses them based on a weighting mechanism [Wang et al., 2019]. Multi-view spectral clustering network [Huang et al., 2019] learns a discriminative representation by using a deep metric learning network. These methods are developed for feature matrix and can not handle graph data. To directly process graph data, some representative methods have also been proposed.

---

[*]Corresponding author.

Scalable multiplex network embedding (MNE) [Zhang et al., 2018] is a scalable multi-view network embedding model, which learns multiple relations by a unified network embedding framework. Principled multilayer network embedding (PMNE) [Liu et al., 2017] proposes three strategies ("network aggregation", "results aggregation", and "layer co-analysis") to project a multilayer network into a continuous vector space. Nevertheless, they fail to explore the feature information [Lin and Kang, 2021].

Recently, based on GCN, One2Multi graph autoencoder clustering (O2MA) framework [Fan et al., 2020] and multi-view attribute GCNs for clustering (MAGCN) [Cheng et al., 2020] achieve superior performance on graph clustering. O2MA introduces a graph autoencoder to learn node embeddings based on one informative graph and reconstruct multiple graphs. However, the shared feature representation of multiple graphs could be incomplete because O2MA only takes into account the informative view selected by modularity. MAGCN exploits the abundant information of all views and adopts a cross-view consensus learning by enforcing the representations of different views to be as similar as possible. Nevertheless, O2MA targets for multiple graphs while MAGCN mainly solves graph structured data of multi-view attributes. They are not directly applicable to multiple graphs data with multi-view attributes. Therefore, the research of multi-view graph clustering is at the initial stage and more dedicated efforts are pressingly needed.

In this paper, we propose a generic framework of clustering on attributed graph data with multi-view features and multiple topological graphs, denoted by *Multi-view Contrastive Graph Clustering* (MCGC). To be exact, MCGC learns a new consensus graph by exploring the holistic information among various attributes and graphs rather than utilizing the initial graph. The reason of introducing graph learning is that the initial graph is often noisy or incomplete, which leads to suboptimal solutions [Chen et al., 2020b, Kang et al., 2020b]. A contrastive loss is adopted as regularization to make the consensus graph clustering-friendly. Moreover, we implement on the smooth representation rather than raw data. The contributions of this work could be summarized as follows:

- To boost the quality of learned graph, we propose a novel contrastive loss at graph-level. It is capable of drawing similar nodes close and pushing those dissimilar ones apart.

- We propose a generic clustering framework to handle multilayer graphs with multi-view attributes, which contains graph filtering, graph learning, and graph contrastive components. The graph filtering is simple and efficient to obtain a smoothed representation; the graph learning is utilized to generate the consensus graph with an adaptive weighting mechanism for different views.

- Our method achieves state-of-the-art performance compared with shallow methods and deep methods on five benchmark datasets.

## 2 Related Work

### 2.1 Multi-view Clustering

Large quantities of multi-view clustering methods have been proposed in the last decades. Multi-view low-rank sparse subspace clustering [Brbić and Kopriva, 2018] obtains a joint subspace representation across all views by learning an affinity matrix constrained by sparsity and low-rank constraint. Cross-view matching clustering (COMIC) [Peng et al., 2019] enforces the graphs to be as similar as possible instead of the representation in the latent space. Robust multi-view spectral clustering (RMSC) [Xia et al., 2014] uses a shared low-rank transition probability matrix derived from each single view as input to the standard Markov chain method for clustering. These methods are designed for feature matrix and try to learn a graph from data. To directly cluster multiple graphs, self-weighted multi-view clustering (SwMC) [Nie et al., 2017] method learns a shared graph from multiple graphs by using a novel weighting strategy. Above methods are suitable for graph or feature data only, and can not simultaneously explore attributes and graph structure. As previously discussed, O2MA [Fan et al., 2020] and MAGCN [Cheng et al., 2020] can handle attributed graph, but they are not direct applicable to generic multi-view graph data.

### 2.2 Contrastive Clustering

Due to its impressive performance in many tasks, contrastive learning has become the most hot topic in unsupervised learning. Its motivation is to maximize the similarity of positive pairs and

distance of negative pairs [Hadsell et al., 2006]. Generally, the positive pair are composed of data augmentations of the same instance while those of different instances are regarded as negatives. Several loss functions have been proposed, such as the triplet loss [Chopra et al., 2005], the noise contrastive estimation (NCE) loss [Gutmann and Hyvärinen, 2010], the normalized temperature-scaled cross entropy loss (NT-Xent) [Chen et al., 2020a]. Deep robust clustering turns maximizing mutual information into minimizing contrastive loss and achieves significant improvement after applying contrastive learning to decrease intra-class variance [Zhong et al., 2020]. Contrastive clustering develops a dual contrastive learning framework, which conducts contrastive learning at instance-level as well as cluster-level [Li et al., 2021]. As a result, it produces a representation that facilitates the downstream clustering task. Unfortunately, theses method can only handle single-view data.

Recently, by combining reconstruction, cross-view contrastive learning, and cross-view dual prediction, incomplete multi-view clustering via contrastive prediction (COMPLETER) [Lin et al., 2021a] performs data recovery and consistency learning of incomplete multi-view data simultaneously. It also obtains promising performance on complete multi-view data. However, it can not deal with graph data. On the other hand, contrastive multi-view representation learning on graphs (MVGRL) [Hassani and Khasahmadi, 2020] method is proposed, which performs representation learning by contrasting two diffusion matrices transformed from the adjacency matrix. It reports better performance than variational GAE (VGAE) [Kipf and Welling, 2016], marginalized GAE (MGAE) [Wang et al., 2017], adversarially regularized GAE (ARGA) and VGAE (ARVGA) [Pan et al., 2018], and GALA [Park et al., 2019]. Different from MVGRL, our contrastive regularizer is directly applied on learned graph.

## 3 Methodology

### 3.1 Notation

Define the multi-view graph data as $G = \{\mathcal{V}, E_1, ..., E_V, X^1, ..., X^V\}$, where $\mathcal{V}$ represents the sets of $N$ nodes, $e_{ij} \in E_v$ denotes the relationship between node $i$ and node $j$ in the $v$-th view, $X^v = \{x_1^v, ..., x_N^v\}^\top$ is the feature matrix. Adjacency matrices $\{\widetilde{A}^v\}_{v=1}^V$ characterize the initial graph structure. $\{D^v\}_{v=1}^V$ represent the degree matrices in various views. The normalized adjacency matrix $A^v = (D^v)^{-\frac{1}{2}}(\widetilde{A}^v + I)(D^v)^{-\frac{1}{2}}$ and the corresponding graph laplacian $L^v = I - A^v$.

### 3.2 Graph Filtering

A feature matrix $X \in \mathbb{R}^{N \times d}$ of $N$ nodes can be treated as $d$ $N$-dimensional graph signals. A natural signal should be smooth on nearby nodes in term of the underlying graph. The smoothed signals $H$ can be achieved by solving the following optimization problem [Zhu et al., 2021, Lin et al., 2021b]

$$\min_H \|H - X\|_F^2 + s \operatorname{Tr}\left(H^\top L H\right),  \tag{1}$$

where $s > 0$ is a balance parameter and $L$ is the laplacian matrix associated with $X$. $H$ can be obtained by taking the derivative of Eq. (1) w.r.t. $H$ and setting it to zero, which yields

$$H = (I + sL)^{-1}X.  \tag{2}$$

To get rid of matrix inversion, we approximate $H$ by its first-order Taylor series expansion, i.e., $H = (I - sL)X$. Generally, $m$-th order graph filtering can be written as

$$H = (I - sL)^m X,  \tag{3}$$

where $m$ is a non-negative integer. Graph filtering can filter out undesirable high-frequency noise while preserving the graph geometric features.

### 3.3 Graph Learning

Since real-world graph is often noisy or incomplete, which will degrade the downstream task performance if it is directly applied. Thus we learn an optimized graph $S$ from the smoothed representation $H$. This can be realized based on the self-expression property of data, i.e., each data point can be represented by a linear combination of other data samples [Lv et al., 2021, Ma et al., 2020]. And the combination coefficients represent the relationships among data points. The objective function on single-view data can be mathematically formulated as

$$\min_S \left\| H^\top - H^\top S \right\|_F^2 + \alpha \|S\|_F^2,  \tag{4}$$

where $S \in \mathbb{R}^{N \times N}$ is the graph matrix and $\alpha > 0$ is the trade-off parameter. The first term is the reconstruction loss and the second term serves as a regularizer to avoid trivial solution. Many other regularizers could also be applied, such as the nuclear norm, sparse $\ell_1$ norm [Kang et al., 2020a]. To tackle multi-view data, we can compute a smooth representation $H^v$ for each view and extend Eq. (4) by introducing a weighting factor to distinguish the contributions of different views

$$\min_{S, \lambda^v} \sum_{v=1}^{V} \lambda^v \left( \left\| H^{v\top} - H^{v\top} S \right\|_F^2 + \alpha \|S\|_F^2 \right) + \sum_{v=1}^{V} (\lambda^v)^\gamma, \tag{5}$$

where $\lambda^v$ is the weight of $v$-th view and $\gamma$ is a smooth parameter. Eq. (5) learns a consensus graph $S$ shared by all views. To learn a more discriminative $S$, we introduce a novel regularizer in this work.

### 3.4 Graph Contrastive Regularizer

Generally, contrastive learning is performed at instance-level and positive/negative pairs are constructed by data augmentation. Most graph contrastive learning methods conduct random corruption on nodes and edges to learn a good node representation. Different from them, each node and its $k$-nearest neighbors ($k$NN) are regarded as positive pairs in this paper. Then, we perform contrastive learning at graph-level by applying a contrastive regularizer on the graph matrix $S$ instead of node features. It can be expressed as

$$\mathcal{J} = \sum_{i=1}^{N} \sum_{j \in \mathbb{N}_i^v} -\log \frac{\exp(S_{ij})}{\sum_{p \neq i}^{N} \exp(S_{ip})}, \tag{6}$$

where $\mathbb{N}_i^v$ represents the $k$-nearest neighbors of node $i$ in $v$-th view. The introduction of Eq. (6) is to draw neighbors close and push non-neighbors apart, so as to boost the quality of graph.

Eventually, our proposed multi-view contrastive graph clustering (MCGC) model can be formulated as

$$\min_{S, \lambda^v} \sum_{v=1}^{V} \lambda^v \left( \left\| H^{v\top} - H^{v\top} S \right\|_F^2 + \alpha \sum_{i=1}^{N} \sum_{j \in \mathbb{N}_i^v} -\log \frac{\exp(S_{ij})}{\sum_{p \neq i}^{N} \exp(S_{ip})} \right) + \sum_{v=1}^{V} (\lambda^v)^\gamma. \tag{7}$$

Different from existing multi-view clustering methods, MCGC explores the holistic information from both multi-view attributes and multiple structural graphs. Furthermore, it constructs a consensus graph from the smooth signal rather than the raw data.

### 3.5 Optimization

There are two groups of variables in Eq. (7) and it's difficult to solve them directly. To optimize them, we adopt an alternating optimization strategy, in which we update one variable and fix all others at each time.

**Fix $\lambda^v$, Update $S$**

Because $\lambda^v$ is fixed, our objective function can be expressed as

$$\min_S \sum_{v=1}^{V} \lambda^v \left( \left\| H^{v\top} - H^{v\top} S \right\|_F^2 + \alpha \sum_{i=1}^{N} \sum_{j \in \mathbb{N}_i^v} -\log \frac{\exp(S_{ij})}{\sum_{p \neq i}^{N} \exp(S_{ip})} \right). \tag{8}$$

$S$ can be elemently solved by gradient descent and its derivative at epoch $t$ can be denoted as

$$\nabla_1^{(t)} + \alpha \nabla_2^{(t)}. \tag{9}$$

The first term is

$$\nabla_1^{(t)} = 2 \sum_{v=1}^{V} \lambda^v \left( -\left[ H^v H^{v\top} \right]_{ij} + \left[ H^v H^{v\top} S^{(t-1)} \right]_{ij} \right). \tag{10}$$

Define $K^{(t-1)} = \sum_{p \neq i}^{N} \exp\left( S_{ip}^{(t-1)} \right)$ and let $n$ be the total number of neighbors (i.e., the neighbors from each graph are all incorporated). Consequently, the second term becomes

$$\nabla_2^{(t)} = \begin{cases} \sum_{v=1}^{V} \lambda^v \left( -1 + \frac{n \exp\left( S_{ij}^{(t-1)} \right)}{K^{(t-1)}} \right), & \text{if } j \text{ in } \mathbb{N}_i^v, \\ \sum_{v=1}^{V} \lambda^v \left( \frac{n \exp\left( S_{ij}^{(t-1)} \right)}{K^{(t-1)}} \right), & \text{otherwise.} \end{cases} \tag{11}$$

Then we adopt Adam optimization strategy [Kingma and Ba, 2015] to update $S$. To increase the speed of convergence, we initialize $S$ with $S^*$, where $S^*$ is the solution of Eq. (5).

**Fix $S$, Update $\lambda^v$**

For each view $v$, we define $M^v = \left\| H^{v\top} - H^{v\top}S \right\|_F^2 + \alpha\mathcal{J}$. Then, the loss function is simplified as

$$\min_{\lambda^v} \sum_{v=1}^{V} \lambda^v M^v + \sum_{v}^{V} (\lambda^v)^\gamma. \tag{12}$$

By setting its derivation to zero, we get

$$\lambda^v = \left( \frac{-M^v}{\gamma} \right)^{\frac{1}{\gamma-1}}. \tag{13}$$

We alternatively update $S$ and $\lambda^v$ until convergence. The complete procedures are outlined in Algorithm 1.

---

**Algorithm 1** MCGC

---

**Require:** adjacency matrix $\widetilde{A}^1,...,\widetilde{A}^V$, feature $X^1,...,X^V$, the order of graph filtering $m$, parameter $\alpha$, $s$ and $\gamma$, the number of clusters $c$.
**Ensure:** $c$ clusters.
1: $\lambda^v = 1$;
2: $A^v = (D^v)^{-\frac{1}{2}}(\widetilde{A}^v + I)(D^v)^{-\frac{1}{2}}$;
3: $L^v = I - A^v$;
4: Graph filtering by Eq. (4) for each view;
5: **while** convergence condition does not meet **do**
6:     Update $S$ in Eq. (9) via Adam;
7:     **for** each view **do**
8:         Update $\lambda^v$ in Eq. (13);
9:     **end for**
10: **end while**
11: $C = \frac{(|S|+|S|^\top)}{2}$;
12: Clustering on $C$.

---

# 4 Experiments

## 4.1 Datasets and Metrics

We evaluate MCGC on five benchmark datasets, ACM, DBLP, IMDB [Fan et al., 2020], Amazon photos and Amazon computers [Shchur et al., 2018]. The statistical information is shown in Table 1.

**ACM:** It is a paper network from the ACM dataset. Node attribute features are the elements of a bag-of-words representing of each paper's keywords. The two graphs are constructed by two types of relationships: "Co-Author" means that two papers are written by the same author and "Co-Subject" suggests that they focus on the same filed.

**DBLP:** It is an author network from the DBLP dataset. Node attribute features are the elements of a bag-of-words representing of each author's keywords. Three graphs are derived from the relationships: "Co-Author", "Co-Conference", and "Co-Term", which indicate that two authors have worked together on papers, published papers at the same conference, and published papers with the same terms.

**IMDB:** It is a movie network from the IMDB dataset. Node attribute features correspond to elements of a bag-of-words representing of each movie. The relationships of being acted by the same actor (Co-actor) and directed by the same director (Co-director) are exploited to construct two graphs.

**Amazon photos and Amazon computers:** They are segments of the Amazon co-purchase network dataset, in which nodes represent goods and features of each good are bag-of-words of product reviews, the edges means that two goods are purchased together. To have multi-view attributes, the second feature matrix is constructed via cartesian product by following [Cheng et al., 2020].

We adopt four popular clustering metrics: Accuracy (ACC), normalized Mutual Information (NMI), Adjusted Rand Index (ARI), F1 score.

Table 1: The statistical information of datasets.

| Dataset | Nodes | Features | Graph and Edges | Clusters |
|---------|-------|----------|-----------------|----------|
| ACM | 3,025 | 1,830 | Co-Subject (29,281) | 3 |
| | | | Co-Author (2,210,761) | |
| DBLP | 4,057 | 334 | Co-Author (11,113) | 4 |
| | | | Co-Conference (5,000,495) | |
| | | | Co-Term (6,776,335) | |
| IMDB | 4,780 | 1,232 | Co-Actor (98,010) | 3 |
| | | | Co-Director (21,018) | |
| Amazon photos | 7,487 | 745 / 7,487 | Co-Purchase(119,043) | 8 |
| Amazon computers | 13,381 | 767 / 13,381 | Co-Purchase(245,778) | 10 |

## 4.2 Experiment Setup

We compare MCGC with multi-view methods as well as single-view methods. LINE [Tang et al., 2015] and GAE [Kipf and Welling, 2016] have been chosen as representatives of single-view methods, and we report the best one among the results from all views. Compared multi-view clustering methods include: PMNE [Liu et al., 2017], RMSC [Xia et al., 2014], SwMC [Nie et al., 2017]. PMNE and SwMC only use structural information while RMSC only exploits attribute features. PMNE uses three strategies to project a multilayer network into a continuous vector space, so we select the best result. MCGC is also compared with other methods that not only explore attribute features but also structural information, i.e., O2MA and O2MAC [Fan et al., 2020], MAGCN [Cheng et al., 2020]. In addition, MCGC is compared with COMPLETER [Lin et al., 2021a] and MVGRL [Hassani and Khasahmadi, 2020] that conduct contrastive learning to learn a common representation shared across features of different views and multiple graphs respectively. For an unbiased comparison, we copy part of the results from [Fan et al., 2020]. Since the neighbors of each node on different views could be different, we also examine another strategy: only use the shared neighbors in the contrastive loss term, $\mathbb{N}_i = \bigcap_{v=1}^{V} \mathbb{N}_i^v$. And our method with this approach is marked as MCGC*. During experiments, $k = 10$ is used to select neighbors and $\gamma$ is fixed as $-4$ since we find that it has little influence to the result. According to parameter analysis, we set $m = 2$, $s = 0.5$, and tune $\alpha$. All experiments are conducted on the same machine with the Intel(R) Core(TM) i7-8700 3.20GHz CPU, two GeForce GTX 1080 Ti GPUs and 64GB RAM. The implementation of MCGC is public available [1].

## 4.3 Results

All results are shown in Table 2 and Table 3. Compared with single-view method GAE, MCGC improves ACC by more than 9%, 4%, 19% on ACM, DBLP, IMDB, respectively. Though using deep neural networks, GAE can not explore the complementarity of views. Compared with PMNE, the ACC, NMI, ARI, F1 are boosted by 16%, 20%, 20%, 12% on average. With respect to LINE, RMSC, SwMC, the improvement is more significant. This can be attributed to the exploration of both feature and structure information in MCGC. Although O2MA, O2MAC, and MAGCN capture attributes and structure information, MCGC still outperforms them considerably. Specifically, MCGC improves O2MAC on average by almost 6%, 9%, 11% on ACC, NMI, F1, respectively. With respect to MAGCN, the improvement is more than 20% for all metrics. Compared with contrastive learning-based approaches, our improvement is also impressive. In particular, compared with COMPLETER, the improvement is more than 30% on Amazon datasets, which illustrates that MCGC benefits from the graph structure information. MCGC also enhances the performance of MVGRL by 20%. By comparing the results of MCGC and MCGC*, we can see that the strategy of choosing neighbors does have impact on performance.

---

[1]https://github.com/Panern/MCGC

Table 2: Results on ACM, DBLP, IMDB.

| Method | | LINE | GAE | PMNE | RMSC | SwMC | O2MA | O2MAC | MCGC | MCGC* |
|---|---|---|---|---|---|---|---|---|---|---|
| ACM | ACC | 0.6479 | 0.8216 | 0.6936 | 0.6315 | 0.3831 | 0.888 | 0.9042 | **0.9147** | 0.9055 |
| | NMI | 0.3941 | 0.4914 | 0.4648 | 0.3973 | 0.4709 | 0.6515 | 0.6923 | **0.7126** | 0.6823 |
| | ARI | 0.3433 | 0.5444 | 0.4302 | 0.3312 | 0.0838 | 0.6987 | 0.7394 | **0.7627** | 0.7385 |
| | F1 | 0.6594 | 0.8225 | 0.6955 | 0.5746 | 0.018 | 0.8894 | 0.9053 | **0.9155** | 0.9062 |
| DBLP | ACC | 0.8689 | 0.8859 | 0.7925 | 0.8994 | 0.3253 | 0.904 | 0.9074 | **0.9298** | 0.9162 |
| | NMI | 0.6676 | 0.6925 | 0.5914 | 0.7111 | 0.019 | 0.7257 | 0.7287 | **0.8302** | 0.7490 |
| | ARI | 0.6988 | 0.741 | 0.5265 | 0.7647 | 0.0159 | 0.7705 | 0.778 | 0.7746 | **0.7995** |
| | F1 | 0.8546 | 0.8743 | 0.7966 | 0.8248 | 0.2808 | 0.8976 | 0.9013 | **0.9252** | 0.9112 |
| IMDB | ACC | 0.4268 | 0.4298 | 0.4958 | 0.2702 | 0.2453 | 0.4697 | 0.4502 | **0.6182** | 0.6113 |
| | NMI | 0.0031 | 0.0402 | 0.0359 | 0.3775 | 0.0023 | 0.0524 | 0.0421 | 0.1149 | **0.1225** |
| | ARI | -0.009 | 0.0473 | 0.0366 | 0.0054 | 0.0017 | 0.0753 | 0.0564 | **0.1833** | 0.1811 |
| | F1 | 0.287 | 0.4062 | 0.3906 | 0.0018 | 0.3164 | 0.4229 | 0.1459 | 0.4401 | **0.4512** |

Table 3: Results on Amazon photos and Amazon computers. The '-' means that the method raises out-of-memory problem.

| Dataset | Amazon photos | | | | Amazon computers | | | |
|---|---|---|---|---|---|---|---|---|
| | ACC | NMI | ARI | F1 | ACC | NMI | ARI | F1 |
| COMPLETER | 0.3678 | 0.2606 | 0.0759 | 0.3067 | 0.2417 | 0.1562 | 0.0536 | 0.1601 |
| MVGRL | 0.5054 | 0.4331 | 0.2379 | 0.4599 | 0.2450 | 0.1012 | 0.0553 | 0.1706 |
| MAGCN | 0.5167 | 0.3897 | 0.2401 | 0.4736 | – | – | – | – |
| MCGC | **0.7164** | **0.6154** | **0.4323** | **0.6864** | **0.5967** | **0.5317** | **0.3902** | **0.5204** |

# 5 Ablation Study

## 5.1 The Effect of Contrastive Loss

By employing the contrastive regularizer, our method pulls neighbors into the same cluster, which decreases intra-cluster variance. To see its effect, we replace $\mathcal{J}$ with a Frobenius term, i.e. Eq. (5). As can be seen from Table 4, the performance falls precipitously without contrastive loss on all datasets. MCGC achieves ACC improvements by 16%, 8%, 5%, 12% on DBLP, ACM, IMDB, Amazon datasets, respectively. For other metrics, the contrastive regularizer also enhances the performance significantly. Above facts validate that MCGC benefits from the graph contrastive loss.

Table 4: Results of MCGC without contrastive loss.

| | Datasets | ACM | DBLP | IMDB | Amazon photos | Amazon computers |
|---|---|---|---|---|---|---|
| ACC | MCGC | **0.9147** | **0.9298** | **0.6182** | **0.7164** | **0.5967** |
| | MCGC w/o $\mathcal{J}$ | 0.8334 | 0.7658 | 0.5636 | 0.5882 | 0.4662 |
| NMI | MCGC | **0.7126** | **0.8302** | **0.1149** | **0.6154** | **0.5317** |
| | MCGC w/o $\mathcal{J}$ | 0.5264 | 0.4621 | 0.0707 | 0.5372 | 0.3988 |
| ARI | MCGC | **0.7627** | **0.7746** | **0.1833** | **0.4323** | **0.3902** |
| | MCGC w/o $\mathcal{J}$ | 0.5779 | 0.4949 | 0.1451 | 0.2640 | 0.1745 |
| F1 | MCGC | **0.9155** | **0.9252** | 0.4401 | **0.6864** | **0.5204** |
| | MCGC w/o $\mathcal{J}$ | 0.8313 | 0.7601 | **0.4444** | 0.5437 | 0.3678 |

Table 5: Results in various views of MCGC on ACM and Amazon photos. $G_1$ and $G_2$ denote the graphs in different views.

| Dataset | ACM | | | Amazon photos | | |
|---|---|---|---|---|---|---|
| | $G_1, X$ | $G_2, X$ | $G_1, G_2, X$ | $X^1, G$ | $X^2, G$ | $X^1, X^2, G$ |
| ACC | 0.9088 | 0.8152 | **0.9147** | 0.4433 | 0.6935 | **0.7164** |
| NMI | 0.6929 | 0.4656 | **0.7126** | 0.3519 | 0.5976 | **0.6154** |
| ARI | 0.7470 | 0.5229 | **0.7627** | 0.1572 | 0.4291 | **0.4323** |
| F1 | 0.9097 | 0.8184 | **0.9155** | 0.3675 | 0.6734 | **0.6864** |

## 5.2 The Effect of Multi-View Learning

To demonstrate the effect of multi-view learning, we evaluate the performance of the following single-view model

$$\min_S \ \left\| H^\top - H^\top S \right\|_F^2 + \alpha \sum_{i=1}^N \sum_{j \in \mathbb{N}_i} - \log \frac{\exp(S_{ij})}{\sum\limits_{p \neq i}^N \exp(S_{ip})}. \tag{14}$$

Taking ACM and Amazon photos as examples, we report the clustering performance of various scenarios in Table 5. We can observe that the best performance is always achieved when all views are incorporated. In addition, we also see that the performance varies a lot for different views. This justifies the necessity of $\lambda^v$ in Eq. (7). Therefore, it is beneficial to explore the complementarity of multi-view information.

## 5.3 The Effect of Graph Filtering

To understand the contribution of graph filtering, we conduct another group of experiments. Without graph filtering, our objective function becomes

$$\min_{S, \lambda^\nu} \sum_{v=1}^V \lambda^v \left( \left\| X^{v\top} - X^{v\top} S \right\|_F^2 + \alpha \sum_{i=1}^N \sum_{j \in \mathbb{N}_i^v} - \log \frac{\exp(S_{ij})}{\sum_{p \neq i}^N \exp(S_{ip})} \right) + \sum_{v=1}^V (\lambda^v)^\gamma. \tag{15}$$

We denote this model as MCGC-. The results of MCGC- are shown in Table 6. With respect to MCGC, ACC on ACM, DBLP, IMDB drops by 0.8%, 1.3%, 0.8%, respectively. This indicates that graph filtering makes a positive impact on our model. For other metrics, MCGC also outperforms MCGC- in most cases.

Table 6: The results of MCGC- (without graph filtering).

| Dataset | ACM | | DBLP | | IMDB | |
|---|---|---|---|---|---|---|
| | MCGC | MCGC- | MCGC | MCGC- | MCGC | MCGC- |
| ACC | **0.9147** | 0.9061 | **0.9298** | 0.9162 | **0.6182** | 0.6109 |
| NMI | **0.7126** | 0.6974 | **0.8302** | 0.7490 | 0.1149 | **0.1219** |
| ARI | **0.7627** | 0.7439 | 0.7746 | **0.7995** | **0.1833** | 0.1804 |
| F1 | **0.9155** | 0.9057 | **0.9252** | 0.9112 | 0.4401 | **0.4509** |

## 6 Parameter Analysis

Firstly, two parameters $m$ and $s$ are applied in graph filtering. Taking ACM as an example, we show their influence on performance by setting $m = [1, 2, 3, 4, 5]$, $s = [0.01, 0.1, 0.3, 0.5, 1, 3, 5, 10]$ in Fig. 1. It can be seen that MCGC achieves a reasonable performance for a small $m$ and $s$.

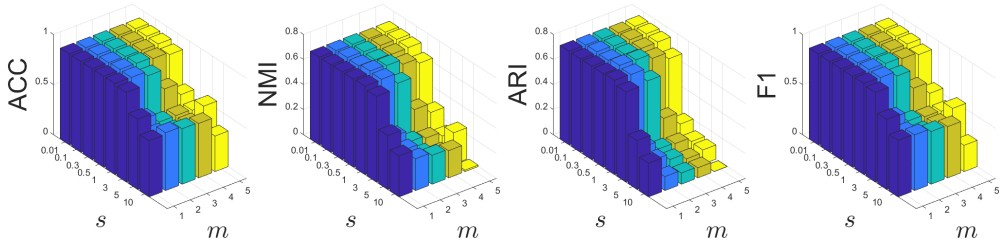

Figure 1: Sensitivity analysis of parameters $m$ and $s$ on ACM.

Therefore, we set $m = 2$ and $s = 0.5$ in all experiments. Afterwards, we tune the trade-off parameter $\alpha = [10^{-3}, 0.1, 1, 10, 10^2, 10^3]$. As shown in Fig. 2, our method is not sensitive to $\alpha$, which enhances the practicality in real-world applications. In addition, we plot the objective variation of Eq. (7) in Fig. 3. As observed from this figure, our method converges quickly.

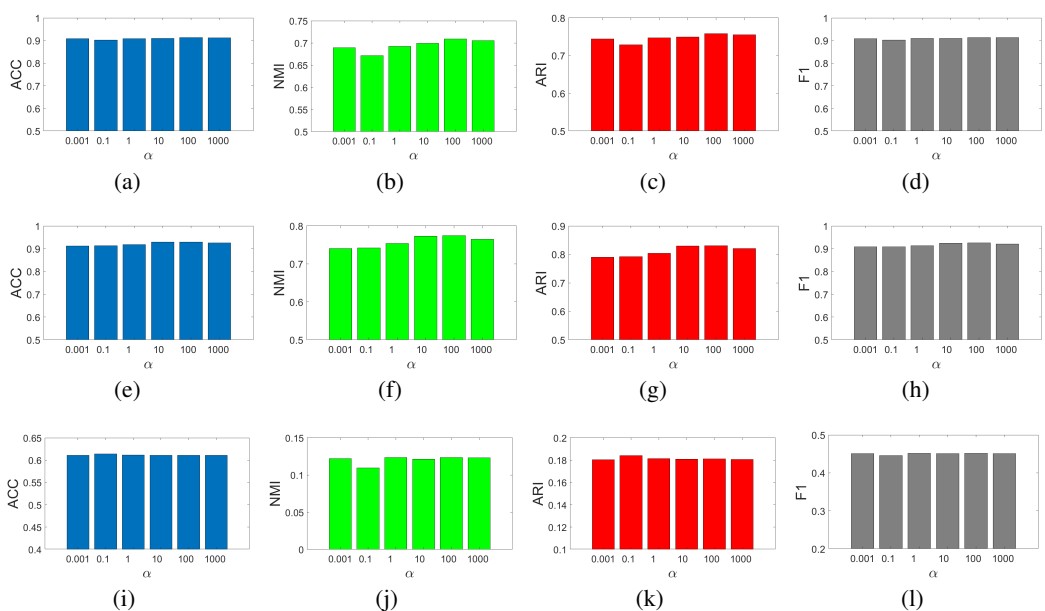

Figure 2: Sensitivity analysis of parameter $\alpha$ on ACM (a-d), DBLP (e-h), IMDB (i-l).

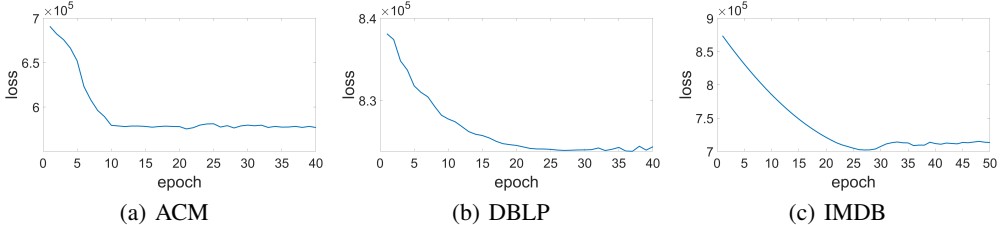

Figure 3: The evolution of objective function.

## 7  Conclusion

Multi-view graph clustering is till at a nascent stage with many challenges remained unsolved. In this paper, we propose a novel method (MCGC) to learn a consensus graph by exploiting not only attribute content but also graph structure information. Particularly, graph filtering is introduced to

filter out noisy components and a contrastive regularizer is employed to further enhance the quality of learned graph. Experimental results on multi-view attributed graph datasets have shown the superior performance of our method. This study demonstrates that it is possible for shallow model to beat deep learning methods facing the systematic use of complex deep neural networks. Graph learning is crucial to more and more tasks and applications. Just like other methods that learn from data, brings the risk of learning biases and perpetuating them in the form of decisions. Thus our method should be deployed with careful consideration of any potential underlying biases in the data. One potential limitation of our approach is that it could take a lot of memory if the data contain too many nodes. This is because the size of learned graph is $N \times N$. Research on large scale network is left for future work.

## Acknowledgments and Disclosure of Funding

This paper was in part supported by the Natural Science Foundation of China (Nos.61806045, U19A2059).

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
