# OpenReview forum: "Multi-view Contrastive Graph Clustering"
_NeurIPS.cc/2021/Conference — NeurIPS 2021 Poster_

### Official Review · Reviewer_fzju · 2021-07-12

**Rating:** 5
**Confidence:** 5

**Summary:**

propose a generic framework to cluster multi-view attributed graph data and borrow the contrastive learning idea to provide a  multi-view contrastive graph clustering (MCGC) method which outpeerforms SOTAs including deep methods.

**Limitations And Societal Impact:**

The authors have pointed out some biases from the data like other methods and remind needing care when deploying the method.

**Main Review:**

Indeed, the submission has good readability and is easy to follow.  proposed MCGC is relatively intuitive and empirically-proved as well effective to some extent,  however, I think its novelty is NOT that significant, using the contrastive learning has had and NOT so unique. In addition, my comments are
1. First there have existed many multi-view clustering (MVC) methods omitted here, secondly, major differences between the traditional MVC and the method for graphs here are unclearly made, especially what challenges will be posed or encountered.
2. On lines 108-109, N-by-d feature matrix X is treated as d "N-dimensional graph signals", obviously, as the N increases, the graph size will be increased accordingly, which seems to bring an impracticality!
3. On lines 115-117, A problem with (3) is if the L is defined by the feature matrix X whose noise will unavoidably propagate into the L, thus H is still noisy unless the L is robustly and jointly learned as well. However, (1) will fail to yield a robust L!
4. On line 143, (7) is just a mixed objective of existing related methods! its optimization is realized via existing techniques.

**Time Spent Reviewing:**

1.5

---

> ### Author Response · Authors · 2021-08-08
> **More details about our paper.**
>
> Thanks for your thoughtful feedback.\
> **Q1**: First there have existed many multi-view clustering (MVC) methods omitted here, secondly, major differences between the traditional MVC and the method for graphs here are unclearly made, especially what challenges will be posed or encountered.\
> **A1**: We can not agree that since we have listed more than *10 MVC methods* including traditional MVC(*RMSC,SwMC,COMIC*), MVC for graph(*MNE,PMNE,O2MA,MAGC*), and MVC with contrastive learning(*COMPLETER,MVGRL*). They are representatives of MVC methods and achieve encouraging performance. In fact, MVC methods that can simultaneously deal with multi-view attributes and multiple graphs are not available, thus it is unnecessary to cite all not closely-related methods. Our work tries to fill the gap. \
> **The major differences between the traditional MVC and the method for graphs**: the traditional MVC methods are developed for feature matrices while the methods for graphs are designed for topological graphs or networks, thus they are designed to process different types of data. As a result, the methods could be totally different. Though there are many algorithms for feature and graph data, methods for attributed multi-view graph data are scarce. The challenge lies in the fusion of both attribute and topology information. \
> **Q2**: On lines 108-109, N-by-d feature matrix X is treated as d "N-dimensional graph signals", obviously, as the N increases, the graph size will be increased accordingly, which seems to bring an impracticality!\
> **A2**: Thanks for raising this concern. N represents the number of nodes. Just like many GNN-based methods, our current model indeed faces the large-scale challenge. As discussed in conclusion, our future work will focus on this problem. \
> In fact, compared to deep neural networks based methods, our method is very simple and efficient. As shown in Table 1, our graph data are already much bigger than that of relevant methods. According to Table 3, MAGCN method can not process Amazon computer data.\
> **Q3**: On lines 115-117, A problem with (3) is if the L is defined by the feature matrix X whose noise will unavoidably propagate into the L, thus H is still noisy unless the L is robustly and jointly learned as well. However, (1) will fail to yield a robust L!\
> **A3**. We want to clarify that L is not defined by the feature matrix X. Instead, L is derived from the adjacency matrix (section 3.1, lines 104-106) .\
> **Q4**: On line 143, (7) is just a mixed objective of existing related methods! its optimization is realized via existing techniques.\
> **A4**. Firstly, we disagree with you that Eq. (7) is just a mixture. On one hand, most existing contrastive methods perform at instance-level and explicitly construct positive/negative pairs by augmentation. Our contrastive regularizer Eq. (6) is defined at graph-level and applied on learned graph directly. We don’t need to perform graph augmentation and we define positive pairs by neighbors. On the other hand, Eq. (7) is constructed for shallow machine learning. Both *Reviewer njps* and *Reviewer gTUK* agree that the graph filtering can be applied to obtain data representation and contrastive learning is a novel way to learn graph. In the era of deep learning, we believe that our research is worth sharing with the community. \
> Secondly, though our optimization is realized via existing techniques, we extend the scope of contrastive learning to shallow machine learning methods.

---

### Official Review · Reviewer_gTUK · 2021-07-15

**Rating:** 6
**Confidence:** 5

**Summary:**

This paper proposes a generic framework for clustering multilayer/multi-view graph data. The main contributions include:
- A graph filtering technique is developed for high-frequency noise removal;
- A contrastive regularizer is applied to graph learning in a simple way;
- Solid experiments and state-of-the-art performance evaluated on five graph data.

**Limitations And Societal Impact:**

The authors have discussed the limitation and potential negative impact of the proposed method.

**Main Review:**

Clarity: The paper is easy to follow and well organized.

Reproducibility: Yes, the source code is publicly available.

Strengths：
1. The challenge tackled by this paper seems interesting and the paper has a good motivation. To my best knowledge, the clustering of multi-view graphs is few studied and this is the first attempt to simultaneously handle multilayer graphs with multi-view attributes data.
2. The proposed method is interesting and novel.  Instead of deep neural networks, the authors use graph filtering to obtain a good representation, which is simple and effective. Moreover, the idea of contrastive learning is employed in a new way to learn graphs in this paper. Though simple, the techniques make sense and look novel to me.
3. The experimental evaluation is comprehensive and the results obtained with the proposed method are convincing. In particular, the authors compare many recent methods from different categories.
4. This work could inspire more researchers/practitioners to apply a contrastive regularizer in shallow machine learning methods.

Weaknesses：
1. As shown in Table 1, the authors conduct experiments with multilayer graphs (ACM, DBLP, IMDB) and multi-view attributes (Amazon photos and computers) respectively. Is it possible to evaluate data with multilayer graph and multi-view features?
2. The contributions of this paper should be explicitly highlighted in the Introduction.
3. This paper presents abundant results with five tables. However, the discussion is quite short and should be elaborated in detail.
4. The authors say that the cartesian product is applied to construct the 2nd feature. However, its definition is unclear.
5. The authors should carefully proofread the whole paper.


**Time Spent Reviewing:**

5

---

> ### Author Response · Authors · 2021-08-08
> **More details about our paper.**
>
> Thank you for your thoughtful and positive feedback. We’re encouraged that you find MCGC simple and useful. We make efforts to simultaneously handle multi-view attributes and multiple graphs via graph filtering and contrastive regularizer at graph-level. Here are our replies to some weaknesses.\
> **Q1**: As shown in Table 1, the authors conduct experiments with multilayer graphs (ACM, DBLP, IMDB) and multi-view attributes (Amazon photos and computers) respectively. Is it possible to evaluate data with multilayer graphs and multi-view features?\
> **A1**: It is a good point. Unfortunately, such datasets are not publicly available. Therefore, we conduct two groups of experiments in Tables 2 and 3 respectively, which demonstrate our method achieves promising performance on multilayer graph as well as multi-view attributes data.\
> **Q2**: The contributions of this paper should be explicitly highlighted in the Introduction.\
> **A2**: Thanks for your suggestion. In fact, we have summarized our method in the last paragraph of the Introduction. We will highlight our contributions in the final version.\
> Q3: This paper presents abundant results with five tables. However, the discussion is quite short and should be elaborated in detail.\
> **A3**: Tables 2 and 3 are used to show the results on five datasets. Tables 4-6 are designed for ablation study, which is a form of discussion. We will improve this part.\
> **Q4**: The authors say that the cartesian product is applied to construct the 2nd feature. However, its definition is unclear.\
> **A4**: The Cartesian product is defined by XX^T.\
> **Q5**: The authors should carefully proofread the whole paper.\
> **A5**: Thanks for your careful reading. We will carefully proofread the whole paper again.

---

### Official Review · Reviewer_p84e · 2021-07-15

**Rating:** 6
**Confidence:** 5

**Summary:**

This paper proposes a multi-view clustering method. They first use graph filter to generate a new node representation for input, and then do self-expression in a linear combination of various views. Afterwards, they propose a graph contrastive regularizer, which mimics the contrastive learning given the nearest neighbors as positive samples and any others as negative ones. Experiments compared with recent works and ablation studies are presented.

**Limitations And Societal Impact:**

I feel this work has some vital limitations like scalability for large dataset. They can use the matrix based method like years ago but the drawbacks of this kind of formulation should be mentioned and discussed.

**Main Review:**

Originality: The task is not new and the novelty seem lies in the contrastive regularizer.

Quality: My major concerns are as follows.
1. The graph contrastive regularizer, the main contribution of this work, is introduced without strong motivation or any deep insights and only takes up less than a half page. This makes the whole paper weak.
2. From Eq. 6, the graph contrastive regularizer uses the neighbors information. That means this task is not a purely unsupervised task unless the neighbors are obtained from the distance of raw features. If so, from my view, this regularizer is close to spectral regularizer. If the authors insist that this new term contributes a lot, then they are supposed to add it on different optimization baselines to see the according improvements.
3. From my understanding, Eq. 6 is similar to the contrastive loss in Raia 2006 if you simply reformulate the objective. Form this view, the improvement happens when the neighbors are reliable.
4. From the alternative optimization, I can see the proposed method only do full batch update, which is not practical for real-world cases due to non-scalability.
5. The simply used weights are not a good option for missing views or noisy views today. From this aspect, the authors over-claimed the impact of this work.
(1) Raia et al., Dimensionality Reduction by Learning an Invariant Mapping, 2006

Clarity: The submission is clearly written.

Significance: Basically this work is not new for me and is lack of deep understanding. It does improve the current methods from the experimental results while the whole paper is weak.

**Time Spent Reviewing:**

4h

---

> ### Author Response · Authors · 2021-08-08
> **More details about our paper.**
>
> Thanks for your reading and feedback.\
> **Q1**: The task is not new and the novelty seem lies in the contrastive regularizer.\
> **A1**: Clustering of multi-view attributes or multilayer graph data is not new. Their combination could be a challenge for existing methods. To this end, our method attempts to simultaneously explore multiple graphs information and various attributes. *Reviewer gTUK* regards our method as **the first** work to handle multilayer graph along with multi-view attributes. Thus, our novelty not only lies in the contrastive regularizer but also the task.\
> **Q2**: The graph contrastive regularizer, the main contribution of this work, is introduced without strong motivation or any deep insights and only takes up less than a half page. This makes the whole paper weak.\
> **A2**: We restate that the contributions of our work contain the whole framework (graph filtering to obtain representation, graph contrastive regularizer, etc.), the simplicity and effectiveness, which are listed as the strong points by *Reviewer gTUK* and *Reviewer njqs*. Based on the assumption of clustering, similar points should be in the same cluster. Therefore, a clustering-friendly representation of data should improve the clustering performance. Contrastive learning attempts to pull similar points closer. Thus, we introduce graph contrastive regularizer to improve the graph learning. \
> **Q3**: From Eq. 6, the graph contrastive regularizer uses the neighbor’s information. That means this task is not a purely unsupervised task unless the neighbors are obtained from the distance of raw features. If so, from my view, this regularizer is close to spectral regularizer. If the authors insist that this new term contributes a lot, then they are supposed to add it on different optimization baselines to see the according improvements.\
> **A3**: The neighbors are determined based on the distance between nodes, which is calculated based on the features. Therefore, our method is an unsupervised method and there is no connection to spectral regularizer. In fact, to show the contribution of graph contrastive regularizer, we have replaced it with a Frobenius term in Section 5.1. Table 4 shows that the Frobenius regularizer gives inferior performance.\
> **Q4**: From my understanding, Eq. 6 is similar to the contrastive loss in Raia 2006 if you simply reformulate the objective. From this view, the improvement happens when the neighbors are reliable.\
> **A4**: Our formula is totally different from Raia 2006, which just uses l2-norm. In that paper, it aims to learn a good mapping that maps similar points to nearby points and dissimilar points to distant points in the low-dimensional space. To some extent, the ideas share some similarities. \
> **Q5**: From the alternative optimization, I can see the proposed method only do full batch update, which is not practical for real-world cases due to non-scalability.\
> **A5**: The current complexity is quadratic of the number of nodes. We could also try many other optimization techniques to speed up, e.g., stochastic gradient descent. But this is not our focus in this paper and we leave the salability issue for future research.\
> **Q6**: The simply used weights are not a good option for missing views or noisy views today. From this aspect, the authors over-claimed the impact of this work.\
> **A6**: We learn weights for different views and a noisy view will be assigned a lower weight. Therefore, our method is flexible to different datasets.

---

> > ### Comment · Reviewer_p84e · 2021-08-21
> > **comments for authors' response**
> >
> > Thanks for authors' response which basically addresses my concerns. The contributions become clearer to me. The main advantage of the proposed method lies in its capability of exploiting both attributes and topology information in multilayer graph. Specifically, the topology graph is applied to filter out high-frequency noisy. After that, a more informative and reliable graph is attained. The authors also give a plausible explanation of using graph contrastive regularizer. For the revised manuscript, I highly suggest to clearly list out the contributions and novelties. Therefore, I would raise my rating.

---

### Official Review · Reviewer_njqs · 2021-07-24

**Rating:** 7
**Confidence:** 5

**Summary:**

This paper considers the multiview graph clustering problem where the goal is to find a cluster given multiple networks. The paper integrates the self-expression strategy and contrastive learning mechanism into a unified framework to perform graph learning. Besides, graph filtering is designed to recover a clean data in advance. An optimization algorithm is developed to solve the objective function. The paper compares the effectiveness of their method with other deep methods and demonstrates that the proposed method is competitive and promising.

**Limitations And Societal Impact:**

Some concerns and questions:
1. The complexity analysis is missing.
2. Eq.(7) should be explained in more details. For example, it is unclear how to set $\lambda$ in Eq.(7).
3. It would be better to construct some other features and see how the proposed method will work.
4. The paper needs a careful proofreading and some typos should be corrected, e.g. Eq.(11).

**Main Review:**

Some strong points:
1. Facing the systematic usage of complex deep learning, this paper makes an important contribution by showing that shallow model is still possible to beat deep neural networks-based methods.
2. The method is simple to implement and easy to understand. The deployment of graph filtering is extremely interesting and could carry a lot of potential for future progress in the field. The combination of contrastive learning and traditional methods also has a lot of potential.
3. Compared to the SOTA methods, the proposed method demonstrates its superiority on five benchmark datasets. Besides, many ablation experiments are carried out to the effectiveness of each component in the proposed objective function.
4. The paper is well organized and easy to follow.

**Time Spent Reviewing:**

4 hours

---

> ### Author Response · Authors · 2021-08-08
> **More details about our paper.**
>
> Thank you for your insightful and positive feedback. We answer your concerns and questions in the following.\
> **Q1**: The complexity analysis is missing.\
> **A1**: Thanks for your suggestion. The overall complexity of our model is O(N^2). We will lower its complexity in our future work.\
> **Q2**: Eq. (7) should be explained in more details. For example, it is unclear how to set λ in Eq. (7).\
> **A2**: We introduce λ to characterize the importance of different views. The last term is designed to make λ smooth.\
> **Q3**: It would be better to construct some other features and see how the proposed method will work.\
> **A3**: Thanks for your suggestion. The reason we choose the Cartesian product is that our main competitor MAGCN shows that the Cartesian product reports better performance than the Fast Fourier Transform (FFT), Gabor transform, Euler transform. In the future, we could investigate more features.\
> **Q4**: The paper needs a careful proofreading, and some typos should be corrected, e.g., Eq. (11).\
> **A4**: Thanks for your advice.

---

### Official Review · Reviewer_NhsB · 2021-07-30

**Rating:** 6
**Confidence:** 4

**Summary:**

This paper proposes a multi-view clustering model, named MCGC. It utilizes some modern techniques, such as graph filtering, graph learning and contrastive learning, to successfully deal with multiple graphs data with multi-view attributes.

**Limitations And Societal Impact:**

Yes.

**Main Review:**

The main focus is on multiple graphs data with multi-view attributes, which is a lack for traditional multi-view clustering, and in this sense, MCGC is a more general framework compared with other SOTAs. It utilizes graph filtering to smooth node features, graph learning to discard noises and contrastive learning to optimize. And it outperforms other baselines to some degree.

However, here are some drawbacks:
1.	there is a big problem in the organization of the article. In introduction, the authors spend many efforts to describe previous multi-view clustering methods, which is confusing to readers, because the readers probably can not make full sense of these works. Meanwhile, the authors don’t state clearly about their motivation, for example why they should use contrastive learning. It’s difficult for me to follow the story.
2.	The authors should survey some graph learning models in the part of related work.
3.	For equation 1, I suggest the authors cite another paper “Interpreting and Unifying Graph Neural Networks with An Optimization Framework” [WWW 2021], which is also a related work.
4.	For graph matrix S, it is initialized randomly, so its degree of freedom is very high. And the training of S is an inefficient process.


**Time Spent Reviewing:**

6

---

> ### Author Response · Authors · 2021-08-08
> **More details about our paper.**
>
> Thanks for your thoughtful feedback. Here are our replies to your concerns:\
> **Q1**: there is a big problem in the organization of the article. In the introduction, the authors spend many efforts to describe previous multi-view clustering methods, which is confusing to readers, because the readers probably cannot make full sense of these works. Meanwhile, the authors don’t state clearly about their motivation, for example why they should use contrastive learning. It’s difficult for me to follow the story.\
> **A1**: Thanks for your suggestion and we will improve the writing. We describe previous multi-view clustering (MVC) to illustrate the drawbacks of existing methods and indicate the challenges on attributed graph data. Our goal is to develop the first framework to cluster multi-view attributed graph data. Since clustering is an unsupervised task, there is no prior knowledge for graph S. Inspired by the success of contrastive learning, we use Eq.(6) to pull the similar points closer and push non-neighbors away. Consequently, the learned graph S will be high-quality.\
> **Q2**: The authors should survey some graph learning models in the part of related work.\
> **A2**: Thanks for your suggestion. In fact, some MVC methods mentioned in our paper are based on graph learning, like MNE, PMNE, MVGRL, SwMC, RMSC, etc. Though there are some graph learning works in the literature, they mainly target for supervised/semi-supervised tasks.\
> **Q3**: For equation 1, I suggest the authors cite another paper “Interpreting and Unifying Graph Neural Networks with An Optimization Framework” [WWW 2021], which is also a related work.\
> **A3**: Thanks for referring us to this new paper.\
> **Q4**: For graph matrix S, it is initialized randomly, so its degree of freedom is very high. And the training of S is an inefficient process.\
> **A4**: Sorry for this confusion. As stated in line 158, we initialize S with the solution of Eq. (5). Therefore, it is not a random matrix.

---

> > ### Comment · Reviewer_NhsB · 2021-08-20
> > **comments for rebuttal**
> >
> > The authors have addressed some main concerns, and the modifications should be clarified in the revised version.

---

### Decision · Program_Chairs · 2021-09-27

**Decision:**

Accept (Poster)

**Comment:**

This paper focuses on multiview graph representation learning, proposing a graph clustering model and demonstrating its usefulness against  a number of baselines. The reviewers felt the model was well motivated with clear contributions, though there were some concerns about presentations, namely on how the main contributions are presented. However, the reviewers were satisfied with the promised changes: I encourage the authors to do so as this will make the paper more generally clearer in terms of impact (as well as proofreading). Finally, the reviewers were generally excited to see strong performance of a shallow model, and I agree that this is an important result in the context of the currently dominant "bigger is better" models. Therefore, I recommend acceptance.